# The Lyotropic Nature of Halates: An Experimental Study

**DOI:** 10.3390/molecules27238519

**Published:** 2022-12-03

**Authors:** Mert Acar, Duccio Tatini, Barry W. Ninham, Federico Rossi, Nadia Marchettini, Pierandrea Lo Nostro

**Affiliations:** 1Department of Chemistry “Ugo Schiff” and CSGI, University of Florence, 50019 Firenze, Italy; 2Materials Physics (Formerly Department of Applied Mathematics), Research School of Physics, Australian National University, Canberra, ACT 2600, Australia; 3School of Science, University of New South Wales, Northcott Drive, Campbell, Canberra, ACT 2612, Australia; 4Department of Earth, Environmental and Physical Sciences, University of Siena, 53100 Siena, Italy

**Keywords:** Hofmeister series, kosmotropicity, chaotropicity, halates, chlorate, bromate, iodate, polarizability

## Abstract

Unlike halides, where the kosmotropicity decreases from fluoride to iodide, the kosmotropic nature of halates apparently increases from chlorate to iodate, in spite of the lowering in the static ionic polarizability. In this paper, we present an experimental study that confirms the results of previous simulations. The lyotropic nature of aqueous solutions of sodium halates, i.e., NaClO_3_, NaBrO_3_, and NaIO_3_, is investigated through density, conductivity, viscosity, and refractive index measurements as a function of temperature and salt concentration. From the experimental data, we evaluate the activity coefficients and the salt polarizability and assess the anions’ nature in terms of kosmotropicity/chaotropicity. The results clearly indicate that iodate behaves as a kosmotrope, while chlorate is a chaotrope, and bromate shows an intermediate nature. This experimental study confirms that, in the case of halates XO_3_^−^, the kosmotropic–chaotropic ranking reverses with respect to halides. We also discuss and revisit the role of the anion’s polarizability in the interpretation of Hofmeister phenomena.

## 1. Introduction

Specific ion or Hofmeister effects consist of the change of a measurable property induced in a particular system when an electrolyte is added, a change that can be often ranked according to a sequence that is commonly referred to as the “Hofmeister series” [1,2,3].

Hofmeister effects are not accounted for by classical theories of electrolytes, electrochemistry, or colloid and surface science. These theories, developed before quantum mechanics, rely only on electrostatic forces between ions and between ions and surfaces. The series differ from substrate to substrate, depending also on the solvent and on polarity and hydrophobicity of interfaces [4]. The phenomena are observed usually (but not always) when the concentration of the salt is greater than 10 mM, where quantum mechanical forces dominate electrostatics [5]. This concentration threshold is commonly reached everywhere in biology and nearly everywhere else. We recall that, originally, dispersion forces are referred to as electromagnetic fluctuation forces at visible frequencies [6]. However, in the *continuum* solvent model, electromagnetic forces include all fluctuation frequencies, from zero to microwave, including collective dipolar, infrared, visible, far UV, and X-ray regions.

While the inclusion of dispersion with electrostatic forces provides the basis for an inclusive framework to accommodate most ion specific phenomena, the whole story is more complicated, and, as we will later see, hydration is a central player [7].

There are no systems where specific ion effects do not occur, from bulk solutions, pH, buffers, activities, zeta and surface and membrane potentials, ion pumps, enzymatic action, and oscillating reactions [8]; from inorganic to organic and biochemical systems; or from aqueous media to nonaqueous solvents [9,10]. The literature on this topic is vast, and the interested reader can refer to the cited works and references therein [9,10,11,12]. Yet, there is no universal behavior to trace Hofmeister phenomena, i.e., in some cases, the series reverses, and in other cases, some differences in the expected order can be found. Often anions are more effective than cations [9].

In order to attempt to quantify and explain the observed specific ion effects, the experimentalist may find it useful to plot the results as a function of some ion-specific physico-chemical parameters (which we will call *descriptors*) that reflect the nature of the ions, their behavior in hydration, adsorption at interfaces, and more often, in general, in solution, under the (usually omitted) assumption that the contribution of the cation and of the anion are independent and additive. This well-established procedure has two advantages: (i) it allows one to demonstrate and quantify the occurrence of a Hofmeister phenomenon [7,13] and (ii) it helps to trace the mechanism and effect of the investigated ions in a particular case [14]. Among these descriptors, the most common include the ionic static polarizability (*α*) [15], the surface tension molar increment (*k*_1_) [16], the lyotropic number (*N*) [17], the Gibbs free energy and entropy of hydration (Δ_hydr_*G* and Δ_hydr_*S*, respectively) [18,19], the entropy change of water (calculated as the difference between the partial molar entropy of the ion and that of a water molecule surrounded by the other solvent molecules, Δ*S*_II_) [20], and the Jones-Dole viscosity coefficient (*B*_JD_) [21].

More literature references and an extensive discussion on these descriptors can be found in Refs. [7,9].

Each descriptor and other physico-chemical parameters are related to the specific nature of the ions, i.e., to their hydration properties, adhesion to interfaces, and interactions with specific sites. We recall that Hofmeister phenomena occur also in nonaqueous and aprotic solvents, where hydrogen bond clusters do not exist, but van der Waals and quadrupolar interactions play a significant role in setting the solvent structure [22,23,24]. This fact has important consequences on several phenomena, for instance, in the stabilization of a protein’s conformation, solubility, and functionalities, and in industrial fermentation processes [25].

The terms chaotropic and kosmotropic, frequently used in specific ion effect studies, refer to the supposed capability of an ion or a molecule to modify the “water structure” [26]. In fact, according to this hypothesis, when an ion enters a bulk water phase, it first perturbs the hydrogen-bonding network and the structure of water molecules in the liquid state. Then, the powerful ion’s electric field around a small and strongly hydrated kosmotrope will have a great impact on the permanent dipole moment of the surrounding water molecules and force a higher order on local water molecules mainly via charge-dipole interactions. On the other hand, the large and poorly hydrated chaotropes, surrounded by a much weaker electrostatic field, will not offset the original perturbation in the water structure and leave the nearby water molecules more disordered with respect to the pure liquid reference. In other words, the strength of the water-water interactions in the bulk phase can be taken as a reference to distinguish between kosmotropes (where ion-water interactions are stronger than water-water interactions) and chaotropes (where ion-water interactions are weaker than water-water interactions) [9]. This effect is thought to take place also in the case of some neutral molecules, such as sugars [27].

Beyond the terms, the concepts related to kosmotropicity and chaotropicity are still debated in the literature [28], for example, in relation to the salting-in and salting-out effects that salts induce in proteins and other macromolecules in water depending on their concentration [29,30,31].

More recent investigations on cellular activities, on the origin of life on Earth, and on the possibility of extraterrestrial life confirm how strong the implications of these phenomena are [32,33,34,35].

Using the words of Ball and Hallsworth, we can state that “chaotropicity might function as one such empirically defined ‘‘black box’’ term that can help us to classify and organize our thinking while acknowledging that at a deeper, mechanistic level, the story is more complex and not so easily compartmentalized” [36]. In other words, far from implying a real, detectable, and measurable water structure, the terms kosmotrope and chaotrope are to be used as a rule of thumb [36], useful to identify the nature of a solute and describe its effects in a particular system.

Usually, kosmotropic ions possess low polarizabilities due to their high charge density, high surface tension molar increments, very high free energies of hydration, and definitely positive Jones-Dole viscosity coefficients. On the other hand, chaotropes possess large polarizabilities (that implies their electronic clouds are very sensitive to external electric fields), lower surface tension molar increments, small free energies of hydration, and negative Jones-Dole viscosity coefficients.

Concerning halides, their free energy of hydration decreases from F^−^ > Cl^−^ > Br^−^ > I^−^. This trend perfectly reflects the strong kosmotropicity of fluoride and the strong chaotropicity of iodide with chloride and bromide somewhere in between.

On the other hand, the opposite trend is found for the halates, XO_3_^−^, where X = Cl, Br, or I. Based on its polarizability, iodate should behave like thiocyanate or iodide, i.e., like strong chaotropes. Instead, its properties, e.g., the thermodynamic functions of hydration, are typical of a strong kosmotrope [1].

The basic theoretical features of density, viscosity, refractive index, conductivity are reported in Appendix B.

In this paper, we report on the experimental values of density, viscosity, refractive index, and conductivity of sodium halates in water in order to investigate their nature in terms of kosmotropicity vs. chaotropicity and to compare our conclusions with the evidence given by previous computational studies [37,38,39].

Finally, we will revisit and discuss the role of polarizability, one of the most important descriptors of Hofmeister phenomena, in the case of halates.

## 2. Results and Discussion

The experimental results for the three halates at different concentrations will be presented and discussed separately in the following order: density, viscosity, conductivity, and refractive index for the three halates.

### 2.1. Density

The density values at 20 °C, 25 °C, 30 °C, 35 °C, and 40 °C of sodium chlorate, bromate, and iodate in water as a function of the concentration are listed in Appendix A. The plots of *ρ* versus the molal concentration of the three salts at constant temperature are shown in Appendix A. At constant concentration and temperature, the density trend is always
iodate > bromate > chlorate

This suggests a kosmotropic behavior of IO_3_^−^ and chaotropic nature for ClO_3_^−^, with BrO_3_^−^ behaving in an intermediate manner.

The standard partial molar volumes ${\bar{V}}_{2}^{o}$ of each salt solution at 25 °C were calculated according to Equation (A4). They are listed in Table 1 and compared with those published by Millero [40]. The Masson’s equation (Equation (A5)) was used to obtain the empirical $S_{V}^{*}$ coefficients to gain insight on the solute-solute and solute-solvent interactions. In Figure 1, a linear fitting of the apparent molar volumes vs. the square root of the concentration is shown. Each salt has a positive slope, indicating the presence of solute-solute interactions, and with an increasing value of the slope ($S_{V}^{*}$) going from sodium chlorate (black circles) to iodate (red circles).

The $S_{V}^{*}$ for iodate is three times larger than that of chlorate. This occurrence can be ascribed to the formation of ion pairs that, in the former, occur at lower concentration, as the conductivity and viscosity data will confirm (see below). This conclusion is in line with the Law of Matching Water Affinities that predicts the formation of stable ion pairs between ions that possess similar solvation features, i.e., when the cation and the anion are either both kosmotropic or chaotropic [41,42].

The standard electrostrictive molar volumes were calculated according to Equation (A6). For the intrinsic volumes of the ions in first approximation, we used the estimates from Padova [43], obtained by assuming that anions and cations in the solution keep the same coordination number they have in the crystal lattice. The results were normalized by dividing the electrostrictive molar volume by the intrinsic volume (${\bar{V}}_{2,\mathrm{el}}^{o}\left(\%\right)$) in order to compare ions of different sizes [44].

The standard partial molar volumes obtained at 20 °C, 30 °C, 35 °C, and 40 °C are listed in Appendix A. The values of ${\bar{V}}_{2}^{o}$ regularly increase with temperature for all salts.

### 2.2. Viscosity

The viscosity values are listed in Appendix A and plotted in Appendix A. Equation (A8) was used to fit the data as a function of the salt concentration (see Figure 2). The *A* coefficients were calculated according to the Falkenhagen-Vernon equation [45,46]. The extracted fitting parameters are listed in Table 2.

The B_JD_ coefficient increases progressively from sodium chlorate to bromate and iodate, which is consistent with a more kosmotropic nature of iodate, the opposite of that found for halides anions (I^−^ < Br^−^ < Cl^−^ < F^−^), as shown in Table 2 [47].

Considering that the D coefficient is related to the ion-pairing effects that take place at relatively high concentrations of salts, we can conclude that sodium iodate has the highest propensity to form ion pairs with respect to sodium chlorate and bromate (see Table 2). This result will be confirmed by the results obtained from the conductivity measurements (see the next subsection).

### 2.3. Conductivity

Conductivities and molar conductivities of the salt solutions at different concentrations are listed in Appendix A. Appendix A shows the plot of the conductivity κ as a function of the molar concentration for the aqueous solutions of sodium chlorate, bromate, and iodate at 25 °C. As shown in Figure 3, the molar conductivity decreases faster for sodium iodate solutions than for sodium bromate and chlorate solutions.

This behavior is related to the formation of ion pairs, which should be more relevant in the case of sodium iodate. In fact, from the *Λ*/$\sqrt{c_{2}}$ plot (see Appendix A), it appears that the linear dependence of the molar conductivity on the square root of the salt concentration holds until a concentration threshold is reached. After such a value, the conductivity of the solution is described by a more complex formula (Equation (A10)). A possible explanation of this behavior might be related to the formation of ion pairs. Roughly, this threshold is 0.25 M for NaIO_3_, 0.38 M for NaBrO_3_, and 0.49 M for NaClO_3_. This result is in line with the tendency of these salts to form ion pairs and is confirmed by their solubilities in water, approximately 0.454 M for NaIO_3_, 2.412 M for NaBrO_3_, and 9.930 M for NaClO_3_ at 20 °C, as discussed by Collins [48].

The limiting molar conductivities *Λ*^∞^ were calculated from the molar conductivities according to Equation (A10) (see Table 3). From these, we obtained the mean ionic activity coefficient γ_±_ (see Appendix A) from Equation (A9). The data are compared with those available in the literature obtained by different methods, i.e., isopiestic method for sodium chlorate and bromate and vapor-pressure osmometry for sodium iodate (see Appendix A).

The Trusdell-Jones equation (Equation (A11)) was used to fit the mean ionic activity coefficients to extract the linear, ion-specific b parameter, positive for kosmotropes and negative for chaotropes. For chlorate and bromate, b is approximately −0.02 dm^3^/mol, while for iodate it has a positive value of 0.11 dm^3^/mol. As in the case of the viscosity, B_JD_ coefficient:IO_3_^−^ > BrO_3_^−^ > ClO_3_^−^
with a progressive lowering in the kosmotropic character of the ion.

### 2.4. Refractive Index and Polarizability

The refractive index values, measured at 20 °C, are listed in Appendix A. Figure 4 shows the concentration dependence of the refractive index for the three sodium halates in water. On the *x*-axis, the concentration is expressed in g.mL^−1^ because these are the units used in the calculation (Equation (A12)).

The values of α that we extracted from Equation (A13) and listed in Table 4 show that the most polarizable ion is iodate, reflecting the greater number of electrons in iodine and, therefore, the extension and softness of its electronic cloud. A smaller polarizability is usually thought to reflect a strong kosmotropic nature of the ion. This is the case, for instance, for Li^+^ and F^−^ [48]. Within the halide group, α increases significantly from F^−^ to I^−^ because of (1) the increasing number of electrons in the anion and (2) the progressively weaker attraction between the nucleus and the electrons in the external orbitals due to the shielding effect of the more numerous inner electrons. This implies that the electronic cloud in iodide is more expanded (actually the polarizability is expressed in terms of a volume) and softer, i.e., the compactness of the cloud is more sensitive to an external electric field. Finally, the polarizability is a very important ion-specific parameter because it appears in the equations that describe the strength of dispersion (London) and induction (Debye) forces [9]: the larger α, the stronger the interactions between an ion or a molecule and its counterpart. These interactions are always attractive, and given the fact that anions possess larger polarizabilities, anions often (but not always) induce stronger Hofmeister effects than do cations [9].

On the basis of these premises, we conclude here that the meaning and relevance of polarizability in the framework of Hofmeister phenomena need to be revisited. In fact, kosmotropes are usually referred to as ions with low polarizability, large free energy and entropy of hydration, large surface tension molar increments, positive values of the Jones-Dole B coefficient, and small or even negative partial molar volumes (see Table 2 and Table 5). Chaotropes are just the opposite.

The results obtained in this work show that within the halates XO_3_^−^ series, IO_3_^−^ is the most kosmotropic species, and ClO_3_^−^ is the most chaotropic. This is the conclusion that can be drawn on the basis of the density, viscosity, and conductivity data.

Instead, the polarizability of the three salts decreases from iodate to chlorate, conflicting with the common opinion that kosmotropes are supposed to possess lower polarizabilities than chaotropes.

It is not simply a matter of shape or of polyatomic ions, as all halates have a pyramidal structure [38,39] and contain one halogen occupying the pinnacle and three oxygens at the base of the pyramid, with a residual negative charge. Instead, the real significant player is hydration. In fact, Table 5 shows that the main hydration parameters are greater for iodate and smaller for chlorate.

It is important to consider all possible solvation sites in a polyatomic ion to obtain a better picture of the overall behavior of the solutes in the Hofmeister series [39]. Finally, we observe that the polarizability of the anion, obtained from the experimental refractive indices of its aqueous solutions, does not match with its lyotropic nature. Instead, it is the presence of an electron-richer and more polarizable atom, such as iodine, that gives rise to the “cationic character” in these polyatomic ions [38,39].

In the end, the electronegativity difference between oxygen and the halogen atom and the structure of halates result in the formation of an asymmetric charge distribution and, thus, in an internal dipole that eventually modifies the interactions of the ion with the solvent [38,39] and defines its kosmo- or chaotropic nature.

### 2.5. Results from Previous Molecular Dynamics Studies

Molecular dynamics and density functional theory studies, confirmed by multi-edge X-ray adsorption fine structure spectroscopy measurements [38], revealed two strongly hydrated regions in the iodate ion that bear opposite charges: the first is around the iodine atom and bears a formally positive charge, whereas the latter encompasses all oxygen atoms and possesses a formally negative charge [38]. The charge separation is due to the electronegativity difference between I (2.66) and O (3.44).

This particular asymmetry in the charge distribution of the iodate ion is thought to be responsible for the peculiar behavior toward the solvating water molecules. Apparently, the positive region is strongly hydrated by three water molecules with a staggered orientation with respect to the oxygens of IO_3_^−^, whilst approximately nine waters hydrate the negative region where the three oxygens are located (see Figure 5). Moreover, the water molecules that surround the positive region are oriented in the “lone pair” position typical of a hydrated cation, with a tilted water dipole moment to allow for the lone pairs to have a direct interaction with the cation [38].

Another computational study investigated the bromate ion/water system in a quantum density functional theory (DFT) framework, examining the solvation shell structure and dynamics [39]. In this case, the interaction of the water molecules with the positively charged bromine produces only a “shoulder” region in the radial distribution function, and not a well-defined hydration shell, as in the case of iodate. The “shoulder” region of water molecules appears to have a preference for a 120° orientation so that the lone pairs of the water’s oxygens can interact favorably with the positively charged bromine. In general, the dynamics occur faster at the “shoulder” region, while those at the solvation shell region possess a slower dynamic compared with the bulk. Interestingly in the “shoulder” region, water molecules have a slower diffusion compared with the bulk. This was ascribed to the fact that although water molecules have fast escape time scales, once they move close to the oxygens, they form hydrogen bonds and do not move away from the ion [39]. To the best of our knowledge, no molecular dynamics simulations have been performed on the chlorate ion.

Table 5 reports the anion radius, hydration shell thickness, and number of water molecules in the hydration shell, as reported by Marcus [18].

In spite of the larger size of iodine respect to bromine and chloride, the halate ions’ dimension increases the opposite way. In conclusion, iodate is more compact than chlorate. This can also be related to the greater propensity of iodine to establish double bonds with oxygen, a feature that decreases in bromine and chlorine. The number of water molecules in the hydration shell (n in Table 5) reported by Marcus, instead, does not change significantly from one ion to another.

## 3. Materials and Methods

Milli-Q water from Millipore with a resistivity of 18.2 MΩ∙cm and a conductivity of 0.055 μS∙cm^−1^ was used. NaClO_3_, NaBrO_3_, and NaIO_3_ were purchased from Acros Organics (with a declared purity of 99%, 99+%, and 99%, respectively). The solutions were prepared by weighing the salts and water, and the concentrations were expressed in molal units. For data analysis, where needed, the molal concentrations were transformed to molar concentrations using the density values obtained in this work.

Density measurements (±5∙10^−6^ g∙cm^−3^) were conducted with an Anton-Paar^©^ DMA 5000 density meter. All measurements were carried out at five different temperatures: 20 °C, 25 °C, 30 °C, 35 °C, and 40 °C (±0.001 °C) as a function of the salt concentration.

An Ubbelohde viscometer with a capillary diameter of 0.36 ± 0.01 mm from Schott (Mainz, Germany) was used. All measurements were conducted at 25 °C in a water bath equipped with an immersion thermostat Lauda E200 comprising a Pt-100 temperature probe that is used for measuring the actual temperature with an accuracy of ±0.01 °C. Each solution was equilibrated for 30 min before performing the viscosity measurement. The flow time (t) was measured by a stopwatch (±0.01 s). and was converted to the solution viscosity (in cP) by *η* = *Aρt*, as *t* is always larger than 200 s [53]. The A constant was calculated using the tabulated value for pure water at 25 °C (0.89040 cP). The viscosity was determined as a function of the salt concentration at constant temperature for the three sodium halates.

The conductivity meter was purchased from Hach, model senIonTM+ EC7 (Lainate, Italy), which operates with an error lower than 0.1% for the conductivity values and lower than 0.2% for the temperature control.

During the experiments, two different probes were used due to the high difference in conductivity values between MilliQ water and the salt solutions. The probes were also purchased from Hach (models sensIon^TM^+ 50 70 with a range from 0.2 μS/cm to 200 mS/cm, and sensIon^TM^+ 50 71 with a range from 0.05 µS/cm to 30 mS/cm). All measurements were carried out at 25 °C.

An Abbé refractometer model NAR-1T LIQUID from Atago Italia Srl (Milan, Italy) was used for the refractive index measurements (± 0.0002 nD). The Abbé refractometer was connected to a water bath. All measurements were carried out at 20 °C.

## 4. Conclusions

This work, on the basis of the measurements of density, conductivity, refractive index, and viscosity of sodium halates (chlorate, bromate, and iodate) aqueous solutions, pinpoints that iodate is a strong kosmotropic ion, while bromate and chlorate possess a chaotropic nature. This is precisely the reversed trend that the spherical and monoatomic halides show, where iodide is the most chaotropic anion and fluoride is the most kosmotropic.

The salt polarizability, obtained from refractive index data, is larger for iodate and smaller for chlorate. This is a very interesting result, as this parameter is a classic descriptor in specific ion effect studies. In fact, kosmotropes, e.g., fluoride or lithium, possess the lowest values of polarizability, whereas chaotropes, such as iodide or cesium, show the largest values of polarizability. With this work, we show that, at least in the case of halates, this correlation does not hold. These data confirm what previous computational studies concluded [38,39]. A deep analysis of the electronic and structural features of the anions suggests that their lyotropic nature is determined basically by their hydration properties which, in turn, depend on the presence of an internal dipole in the ion due to the different electronegativity and size of the halogen atom.

In the near future, we will address this topic for other series of anions in order to highlight the relevance of their size, shape, and electronegativity in their properties and in the effects they produce in solution.

## Figures and Tables

**Figure 1 molecules-27-08519-f001:**
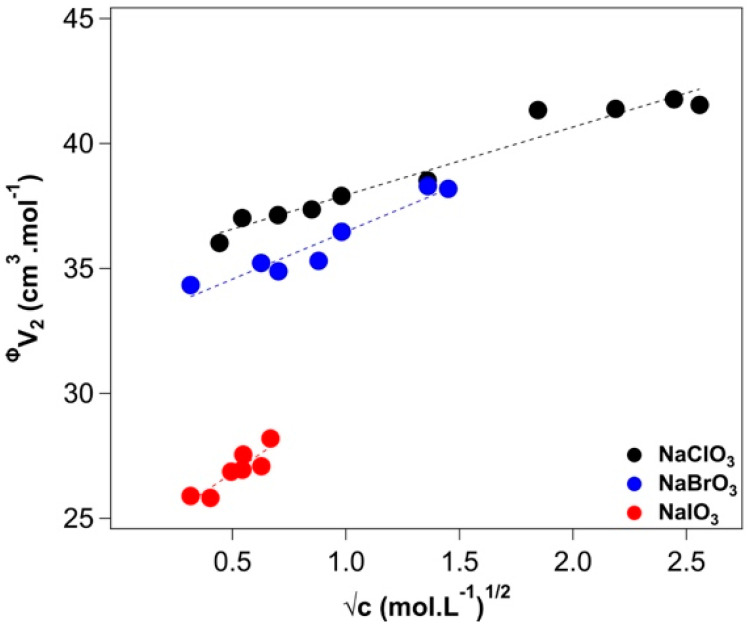
Linear fitting of the Masson’s equation (Equation (A5)). Sodium chlorate (black), bromate (blue), and iodate (red) solutions. The experimental error is ±1%.

**Figure 2 molecules-27-08519-f002:**
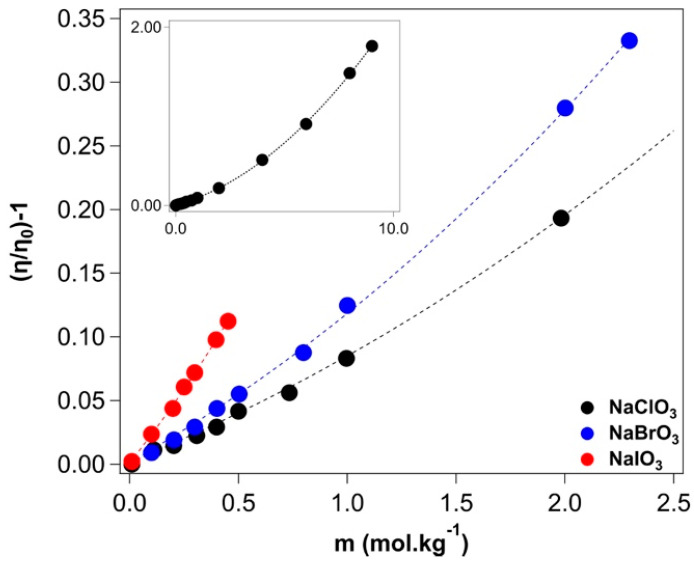
Extended Jones-Dole (Equation (A8)) fitting for sodium chlorate (black), bromate (blue), and iodate (red) solutions. The inset shows the values in the entire range of concentrations for sodium chlorate.

**Figure 3 molecules-27-08519-f003:**
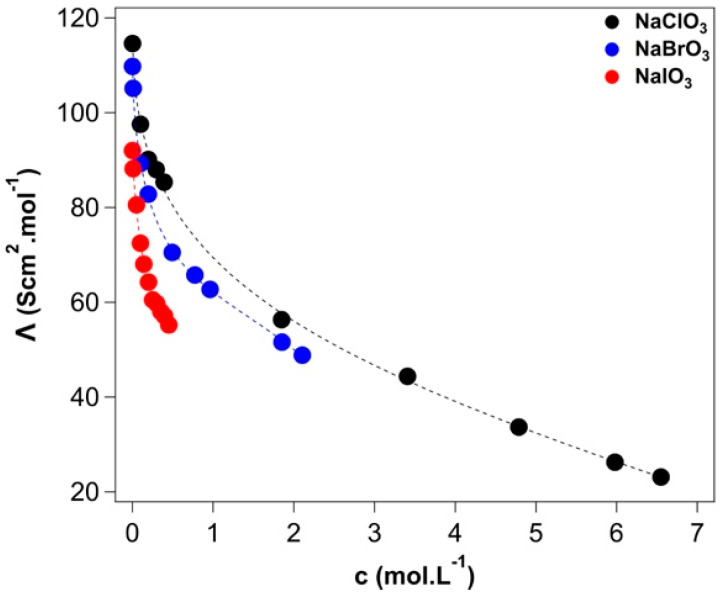
Molar conductivity *Λ* as a function of concentration (c, in molar units) of sodium chlorate (black), bromate (blue), and iodate (red) solutions. Dotted lines represent the fitting of Equation (A10). The experimental absolute error on the molar conductivity values is ±0.1.

**Figure 4 molecules-27-08519-f004:**
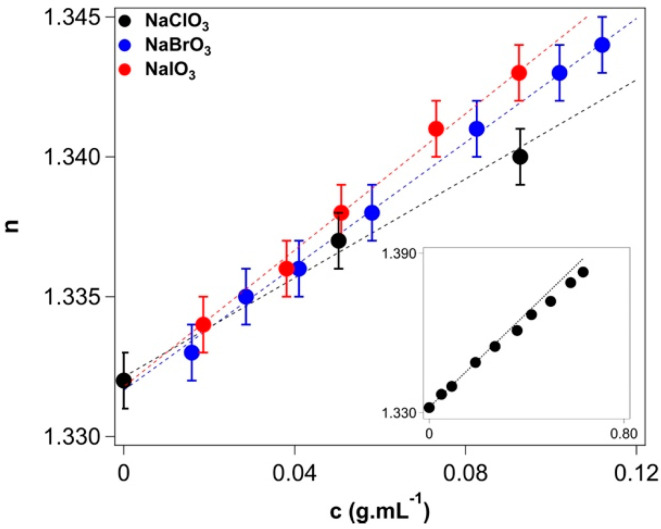
Refractive index values at 20 °C as a function of the concentration of sodium chlorate (black), bromate (blue), and iodate (red) solutions. Dotted lines represent the linear fitting according to Equation (A12). The inset shows the measured refractive index for the entire range of concentration for sodium chlorate investigated in this work.

**Figure 5 molecules-27-08519-f005:**
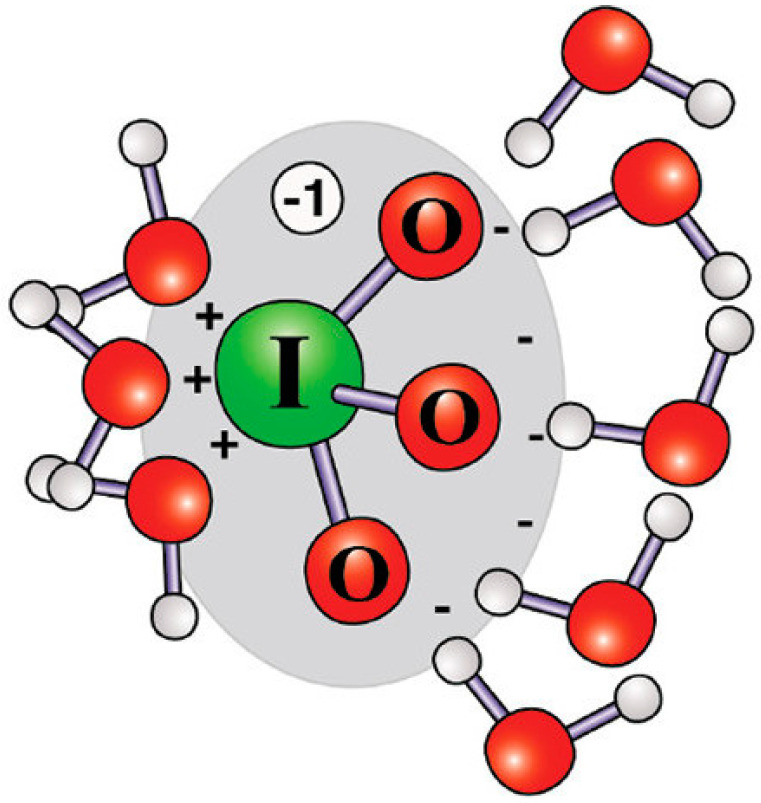
Schematic representation of an IO_3_^−^ ion surrounded by hydrating water molecules in the two regions that bear positive (left) and negative (right) charges. Reprinted with permission from Ref. [38]. Copyright 2011 American Chemical Society.

**Table 1 molecules-27-08519-t001:** Standard partial molar volumes ${\bar{V}}_{2}^{o}$ calculated according to Equation (A4) and compared with the values reported in Ref. [40], standard electrostrictive molar volumes ${\bar{V}}_{2,\mathrm{el}}^{o}$ for each investigated salt at 25 °C expressed in (cm^3^∙mol^−1^) according to Equation (A6), and $S_{V}^{*}$ coefficients obtained by fitting the data according to Equation (A5).

Salt	${\bar{V}}_{2}^{o}$	${\bar{V}}_{2}^{o}$ **[40]**	${\bar{V}}_{2,\mathrm{el}}^{o}$	${\bar{V}}_{2,\mathrm{el}}^{o}\left(\%\right)$	$S_{V}^{*}$
NaClO_3_	35.7 ± 0.2	35.5	−9.4 ± 0.2	−21	2.7 ± 0.2
NaBrO_3_	32.8 ± 0.2	34.1	−14.4 ± 0.2	−31	3.8 ± 0.5
NaIO_3_	24.7 ± 0.3	24.1	−12.1 ± 0.3	−33	6.2 ± 1.3

**Table 2 molecules-27-08519-t002:** *A* (in mol^−1/2^·L^1/2^), *B*_JD_ (in mol^−1^·L), and *D* (in mol^−3/2^·L^3/2^) coefficients obtained by fitting the data with Equation (A8). The *B*_JD_ values are compared with those reported by Ref. [21] and with those of sodium halide [21].

Halate	A	*B* _JD_		D	Halide	*B* _JD_
		This Work	Ref. [21]			Ref. [21]
NaClO_3_	0.0066	0.064 ± 0.002	0.063	0.015 ± 0.001	NaCl	0.080
NaBrO_3_	0.0071	0.089 ± 0.004	0.094	0.023 ± 0.002	NaBr	0.052
NaIO_3_	0.0083	0.197 ± 0.008	0.225	0.089 ± 0.021	NaI	0.012

**Table 3 molecules-27-08519-t003:** Limiting molar conductivity (*Λ*^∞^, in S·cm^2^·mol^−1^) values at 25 °C of sodium chlorate, bromate, and iodate solutions obtained from the fitting of Equation (A10) and from Ref. [49].

Salt	This Work	Ref. [49]
NaClO_3_	116.51 ± 0.74	114.68
NaBrO_3_	112.22 ± 0.36	105.78
NaIO_3_	91.98 ± 0.98	90.58

**Table 4 molecules-27-08519-t004:** Refractive indices (*n*_salt_) obtained from Equation (A12), and polarizabilities (*α*, in Å^3^) obtained from Equation (A13) compared with the literature values.

Salt	n_salt_	α	A ^a^
NaClO_3_	1.553 ± 0.007	5.40 ± 0.06	5.23 (5.43)
NaBrO_3_	1.702 ± 0.007	6.94 ± 0.05	6.47 (6.49)
NaIO_3_	1.854 ± 0.009	8.22 ± 0.07	8.01 (7.64)

^a^ Polarizability values calculated using the experimental values for halate anions from Ref. [50] and for the sodium ion from Ref. [51]. The values in parentheses were calculated using the theoretical polarizability of halate anions from Ref. [52].

**Table 5 molecules-27-08519-t005:** Anion radius (*r*, in nm), hydration shell thickness (Δ*r*, in nm), and number of water molecules in the hydration shell (n). Free energy change (Δ_hydr_*G*) and entropy change (Δ_hydr_*S*) of hydration, lyotropic number (*N*), molar surface tension increment (*k*_1_) and entropy change of water (Δ*S*_II_).

Ion	*r* ^a^	Δ*r* ^a^	*n* ^a^	Δ_hydr_*G* ^a^	Δ_hydr_*S* ^b^	*N* ^c^	*k*_1_ ^d^	ΔS_II_ ^e^
Na^+^	0.102	0.116	3.5	−365	−111	100	1.20	−5.4
ClO_3_^−^	0.200	0.033	1.8	−280	−80	10.65	0.00	5.0
BrO_3_^−^	0.191	0.038	1.9	−330	−95	9.55	0.35	−5.0
IO_3_^−^	0.181	0.043	2.0	−400	−148	6.25	0.70	(−47)

^a^ From Ref. [18]; ^b^ from Ref. [19]; ^c^ from Ref. [17]; ^d^ from Ref. [14]; and ^e^ from Ref. [20].

## Data Availability

The data supporting reported results can be found with the authors.

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
