# Peer review of "The Lyotropic Nature of Halates: An Experimental Study"

_molecules, 2022, doi:10.3390/molecules27238519_

Round 1

Reviewer 1 Report

The authors report profound data on various phyico-chemical properties of aqueous sodium iodate, bromate and chlorate solutions which are associated to the chaotropic or kosmotropic potential of the solutes. Counter-intuitively iodate, which possesses the highest polarizability, exhibits the strongest kosmotropic behavior. The authors explain this with the formation of a dipole, which they back up by references to simulation studies. The study was prepared very carefully and may be interesting to anyone working on ion-specific interactions. However, the author should thoroughly proofread the manuscript again before resubmission to take care of several minor language and formatting issues.

Detailed Remarks:

Lines 33-34: Grammar: “These theories were developed before quantum mechanics and rely only on electrostatic forces between ions..” or “These theories developed before quantum mechanics, rely only on electrostatic forces between ions..”

Line 39-40: Reference needed, also I think “as” is missing in this sentence

Line 43: This sentence is unintelligible and maybe unnecessary, considering that “visible frequencies” are already mentioned two sentences before.

Line 63: “ionic static polarizability ( )” symbol is missing

Lines 64, 65, 67: “DhydrG” Should “D” be the greek Letter Delta? Occurs several times

Line 170: The authors should specify from what the behavior is different.

Line 192: Reference is missing

Section 2.3 does not contain any results from the authors but strictly discusses literature data. The section should be marked accordingly (and maybe even moved to the end of the Results and Discussion section if appropriate)

Line 208: please show the plot in the SI

Line 250 “a” should be in italics (also in Line 262)

Line 337, 338: Does the dipole formation maybe also depend on the size of the halogen atom and how the oxygen atoms can arrange around it? (Just a thought)

Line 308: needless full stop

Line 309: h = Art?

Line 386: I believe this sentence is missing a verb, e.g. “but also depend on”

Equation A2: I believe that this equation only holds true if the solvent is water with “1000” referring to the molar mass of the water. The authors should clarify this and also take care that all necessary units are contained.

Line 474: There is one “increase” to much

Line 507: “in I” à “in equation A11”?

General Remark:

I highly welcome that the authors listed their exact measurements values in the Supplementary Information. This not only substantiates there results, but also provides the data in a reusable manner as required by the FAIR-Data Principles. Nonetheless, I believe that readers would benefit from an additional graphical representation of the data, e.g. for table S1 plotting the density vs. the molality for all three salts in one graph. This would facilitate the comparability between different data sets largely.

Reviewer 2 Report

1.     In general, A brief discussion on the advantage of this study in real-life phenomena and the advancement of science should be discussed.

2.     General discussion on chaotropes and kosmotrope (especially fundamentals) properties can be helpful for readers overall.

3.     The color code for all the figures should be included inside the figure. This can be helpful for the readers.   

4.     The image quality for Figures 2 and 5 must be enhanced for clear visualization.

5.     Most of the citations are very old. New work/reports should be cited more to enhance the quality of this MS.

6.     The future aspects of using this result should be discussed in this MS.

7.     Some typos are there (e.g., in the Title, all initial letters should be in upper cases, etc.).
